# Architecting Nanostructured Co-BTC@GO Composites for Supercapacitor Electrode Application

**DOI:** 10.3390/nano12183234

**Published:** 2022-09-18

**Authors:** Tianen Chen, Allen Yang, Wei Zhang, Jinhui Nie, Tingting Wang, Jianchao Gong, Yuanhao Wang, Yaxiong Ji

**Affiliations:** 1Hoffmann Institute of Advanced Materials, Shenzhen Polytechnic, Shenzhen 518055, China; 2Nord Anglia International School, Hong Kong, China; 3PetroChina Petrochemical Research Institute, Beijing 102206, China

**Keywords:** graphene oxide, metal-organic framework, composite, electrochemical, supercapacitor

## Abstract

Herein, we present an innovative graphene oxide (GO)-induced strategy for synthesizing GO-based metal-organic-framework composites (Co-BTC@GO) for high-performance supercapacitors. 1,3,5-Benzene tricarboxylic acid (BTC) is used as an inexpensive organic ligand for the synthesis of composites. An optimal GO dosage was ascertained by the combined analysis of morphology characterization and electrochemical measurement. The 3D Co-BTC@GO composites display a microsphere morphology similar to that of Co-BTC, indicating the framework effect of Co-BTC on GO dispersion. The Co-BTC@GO composites own a stable interface between the electrolyte and electrodes, as well as a better charge transfer path than pristine GO and Co-BTC. A study was conducted to determine the synergistic effects and electrochemical behavior of GO content on Co-BTC. The highest energy storage performance was achieved for Co-BTC@GO 2 (GO dosage is 0.02 g). The maximum specific capacitance was 1144 F/g at 1 A/g, with an excellent rate capability. After 2000 cycles, Co-BTC@GO 2 maintains outstanding life stability of 88.1%. It is expected that this material will throw light on the development of supercapacitor electrodes that hold good electrochemical properties.

## 1. Introduction

Supercapacitors have attracted a lot of attention as a pollution-free way of storing electrical energy with long cycle life and high power density [1,2,3,4,5]. Recently, they have partially replaced conventional rechargeable batteries in industries such as elevators, electric vehicles, mobile electronics, and rail transportation fields. The emergence of supercapacitors is generally considered to be the most effective power supply system to bridge the gap between low energy density conventional capacitors and low power density lithium-ion batteries [6,7,8].

Graphene oxide (GO), generally considered by researchers as a chemical derivative of graphene with oxygen functional groups, is frequently regarded as an excellent and efficient material for supercapacitors due to its excellent thermal and electrical conductivity, favorable mechanical properties, high electron mobility, and other outstanding properties [9,10,11,12,13]. At the same time, GO has some of the same excellent properties as graphene. Most critically, GO is an ideal material for electrode doping because of its abundant source, very low price, and environmental friendliness [14,15,16,17]. In addition, because GO contains oxygen-containing groups on its surface, it can be used as a surface functionalizer, thereby making it more suitable for integrating with organic substrates and the decoration of functional materials [18,19]. Strong directional interactions between GO sheets, however, often result in GO sheets stacking and reducing its surface area [9,20,21,22,23,24].

Metal-organic frameworks (MOFs) are porous crystal structures that have recently received a lot of attention from researchers because of their regularity, porosity, and large specific surface area, as these properties can increase the rate of electron and ion transfer during electrochemical reactions [25,26,27,28]. Further, MOFs have multiple electron arrangements in their outer electron orbits, which enables multi-level redox behavior and more charge storage possibilities [29,30]. However, pristine MOFs have poor electrical conductivity, so they are rarely used for supercapacitor electrodes. To overcome the disadvantages of MOFs, one approach is to design and synthesize highly conductive MOF-based hybrid composites [31,32,33,34,35,36,37,38]. In one study, CNTs@Co-BTC microspheres (CCBM) were synthesized via solvent heat. Electrochemical test results showed that the MOF microspheres (CCBM) had the better electrical capacity and cycling performance than Co-BTC microspheres (CBM). CCBM apparently possessed a higher specific electric capacity (553.3 F/g at 1 A/g) than CBM (371.8 F/g at 1 A/g) [39]. The other method is to use MOF derivatives as electrode materials for supercapacitors to obtain good conductivity and large specific capacitance. For example, hollow carbon nanofibers with an inside-outside decoration of bi-metallic MOF-derived Ni-Fe phosphides ((Ni-Fe)-P-C@HCNFs) electrode have a high specific capacitance of 1392 F/g at 1 A/g and good cycling stability [40]. It is also possible to construct multicomponent composites using MOFs containing highly conductive materials. For instance, MOF was grown directly on GO sheets and used as a hybrid supercapacitor electrode with a maximum specific capacitance of 447.2 F/g at 1 A/g [41]. In recent years, several groups have recognized the benefits of combining MOF and GO for new special composite materials, not only solving the problem of graphene oxide superimposition caused by the mutual orientation effect, but also improving the conductivity of MOFs [42,43,44,45,46,47]. The composites with outstanding properties can be used in supercapacitor applications.

Herein, we report a facile method combining Co-BTC and GO to fabricate highly porous three-dimensional structures to meet the demand for high-performance supercapacitors. The basic principle is to combine the 2D lamellar structure of GO with the 3D structure of Co-BTC microspheres to produce composites with excellent electron and charge transport properties. The ultimate objective of this study is to analyze the morphological characteristics of the composites and evaluate their electrochemical properties, while the effect of the Co-BTC:GO ratio on capacitance is fully investigated and discussed by synthesizing Co-BTC@GO with different graphene oxide content.

## 2. Experimental

### 2.1. Materias

Cobalt nitrate hexahydrate (Co(NO_3_)_2_·6H_2_O, AR, 99%, Aladdin Inc., Shanghai, China), 1,3,5-benzenetricarboxylic acid (H_3_BTC, 98%, Aladdin Inc., Shanghai, China), and N,N-dimethylformamide (DMF, AR, 99.5%, Macklin Inc., Shanghai, China) and ethanol (AR, >99.7%, Aladdin Inc., Shanghai, China) were used without further purification. Graphite powder (99.8%, Aladdin Inc., Shanghai, China), sodium nitrate (NaNO_3_, 99.99%, Aladdin Inc., Shanghai, China), sulfuric acid (H_2_SO_4_, 98.0%, Guangzhou Chemical Reagent Factory, Guangzhou, China), potassium permanganate (KMnO_4_, AR, >99.5%, Guangzhou Chemical Reagent Factory, Guangzhou, China), hydrogen peroxide (H_2_O_2_, AR, 30.0%, Xilong Scientific, Guangzhou, China), and hydrochloric acid (HCl, AR, 35%, Sinopharm Chemical Reagent Co., Ltd, Shanghai, China) were also used without further purification. Carbon (Macklin Inc. Shanghai, China), polyvinylidene fluoride (PVDF, >99.0%, Sigma-Aldrich Inc., Shanghai, China), Nmethyl-2-pyrrolidone (NMP, AR, >99.0%, Aladdin Inc., Shanghai, China), Nickel foam (1 mm × 200 mm × 300 mm, SuZhou Keshenghe Co., Ltd, Suzhou, China) were also used.

### 2.2. Synthesis of Graphene Oxide

The improved Hummer’s method was applied for preparing GO. Detailed experimental steps were as follows [46]: certain amounts of graphite (1 g) and NaNO_3_ (0.5 g) were mixed in 23 mL of H_2_SO_4_ solution and stirred in the ice bath for 30 min to pre-oxidize. Subsequently, 4 g of KmnO_4_ was added under stirring for 2 h. After temperatures reached 35 °C, the solution was continuously stirred for 30 min, which was then added to 50 mL of DI water. Afterwards, DI water (100 mL) and H_2_O_2_ (30 mL) were added after being reacted for 3 h at 98 °C. The final products were vacuum filtered and rinsed using HCl (5 wt%) and then washed with DI water until pH = 7. GO powder was obtained after freeze-drying overnight.

### 2.3. Synthesis of Co-BTC

Co-BTC was synthesized according to the previously reported literature [47]. Firstly, 1164 mg of Co(NO_3_)_2_·6H_2_O and 291 mg of H_3_BTC were mixed in 29.1 mL of DMF and ultrasounded for 30 min. Afterwards, the solution was transferred to an autoclave lined with Teflon and kept in an oven at 120 °C for 15 h. The final product was obtained by natural cooling, filtration, washing with DMF and ethanol in turn, as well as dried overnight at 80 °C.

### 2.4. Synthesis of Co-BTC@GO

Figure 1 shows the process for the preparation of the Co-BTC@GO composites. Typically, various amounts of GO (0, 0.01, 0.02, and 0.04 g) were respectively dispersed in 29.1 mL of N,N-diethylformamide (DMF) and sonicated for half an hour to obtain homogeneously-dispersed graphene oxide, followed by adding 291 mg of 1,3,5-tricarboxylic acid (H_3_BTC) and 1164 mg cobalt nitrate; then, the mixture solution was stirred for 30 min. The blend was manufactured by the same process as described in Section 2.3. Annotations are based on GO mass: Co-BTC@GO 0, Co-BTC@GO 1, Co-BTC@GO 2, Co-BTC@GO 4 (corresponding to 0, 0.01, 0.02, and 0.04 g of GO, respectively).

### 2.5. Morphological Characterization

Powder X-ray diffraction (XRD; D8 Advance, Bruker, Karlsruhe, Germany) was carried out to investigate the microstructure of the composites acquired with Cu Kα radiation at 35 mA and 40 kV (λ = 1.54 Å) over a 2θ range of 3–50°. Fourier-transform infrared (FTIR, Spectrum GX, PerkinElmer, Waltham, MA, USA) spectroscopic analysis was carried out via the standard KBr disk method in the wavelength range of 400–4000 cm^−1^. Scanning electron microscopy (TESCAN MIRA LMS, Tescan, Brno, Czech Republic) was used to analyze the surface morphology and microstructure of the composites at 200 eV–300 keV. For elemental analysis, an energy-dispersive X-ray spectroscopy detector was coupled to the SEM. Transmission electron microscopy (FEI talos F200× G2, FEI, Hillsboro, OR, USA) further analyzed the microstructure of the composites. For more in-depth evaluation, data for the composites were collected using a photoelectron spectroscopy (Thermo Scientific K-Alpha, Thermo Scientific, Waltham, MA, USA) with monochromated Al-Kα radiation (1486.6 eV) under vacuum.

## 3. Result and Discussion

### 3.1. Morphological Characterization

The XRD patterns of GO, Co-BTC@GO 0, Co-BTC@GO 1, Co-BTC@GO 2, Co-BTC@GO 4 are shown in Figure 2a. The peak at 2θ = 10.1° corresponds to the (001) reflection of GO [41]. Additional diffraction peaks are observed at 2θ = 7.3, 10.9, 12.6, 13.2°, 14.7, 16.7, 18.3, 21.9, 22.3, 23.1 and 25.5° compared to GO in the XRD patterns of the Co-BTC and Co-BTC@GO composites, which show good agreement with previous literature reports [48]. The patterns for Co-BTC@GO 0, Co-BTC@GO 1, and Co-BTC@GO 2 show obvious similarities, whereas some peaks for Co-BTC@GO 4 (peaks at 7.3, 10.1, and 10.9°) are missing. The excessive addition of GO, therefore, has a great effect on the phase and crystallinity of Co-BTC. Moreover, a narrower peak in the XRD pattern means a greater degree of crystallinity due to the fact that MOFs (i.e., Co-BTC in this study) are porous crystal structures, which act as the main phase in MOF@GO-1 composites under low GO dosage. Therefore, from the results of the XRD analysis, it can be concluded that the Co-BTC@GO composite was successfully prepared.

Figure 2b shows the FTIR spectra for GO, Co-BTC@GO 0, Co-BTC@GO 1, Co-BTC@GO 2, and Co-BTC@GO 4 in the range of 400–4000 cm^−1^. The peak at 3408.5 cm^−1^ is due to the existence of -OH groups. In the spectrum of GO, two obvious peaks at 1724.1 and 1619.9 cm^−1^ are related to the C=O stretching vibration and ester groups, which slightly shifted after being functionalized with Co-BTC, indicating the existence of strong interactions between the functional group in the GO and the organic linker in Co-BTC. In the spectra of Co-BTC@GO composites, the obvious peak at 1375.5 cm^−1^ is due to the alcoholic C-OH stretching vibration in the organic linker, BTC. The peaks at 1108.9 and 715.5 cm^−1^ correspond to the C-O-Co bond assignments within Co-BTC. From the results of the IR spectra, it can be concluded that Co-BTC and GO were successfully combined.

The morphology of the as-prepared composites was observed by SEM. Co-BTC crystals are fine microspheres with a smooth surface, the sizes of which are less than 2 μm (Figure 3a). In contrast to Co-BTC, the Co-BTC@GO composites undergo different phase changes with obviously rough surfaces (Figure 3b–d). The composites display a microsphere morphology similar to Co-BTC, indicating the framework effect of Co-BTC on GO dispersion. EDS mapping of C, O, and Co elements in Co-BTC@GO 2 reveals that all constituent elements are well distributed throughout the crystal, verifying the successful dispersion of GO on the Co-BTC surface. Moreover, the existence of the Co element indicates the partial coverage of GO on the particle surface, which could not only enhance the effective contact area between the electrode and electrolyte at the same time as shortening the diffusion path of the ions but can also reserve the Co^2+^ active site throughout the electrochemical test. Under low GO dosage, GO sheets are intercalated into the channels of Co-BTC (Figure 3b,c). Under high GO dosage, however, GO sheets cover the surface of Co-BTC, inducing a different morphology for Co-BTC@GO-4 into Co-BTC@GO-1 and Co-BTC@GO-2. The size of the Co-BTC@GO 4 nanoparticle is similar, but the microporosity is less than that of Co-BTC@GO 1 and Co-BTC@GO 2, which should be attributed to GO intervention limiting MOF crystal nuclei growth. Compared with Co-BTC, Co-BTC@GO-4 is characterized by a similarly smooth surface but a significantly larger crystal size. Compared with Co-BTC@GO-1 and Co-BTC@GO-2, it can be concluded that GO sheets are firstly intercalated, followed by the coverage of the GO sheets over the MOF surface, with GO dosage increasing. Due to the flexibility of the GO sheets, excess GO may induce the aggregation of GO sheets on the MOF surface, resulting in a similar crystal structure of Co-BTC@GO-4 to Co-BTC. This is also supported by the XRD characterization results.

The morphology of the Co-BTC@GO composites was further characterized by TEM. The spherical structure of the Co-BTC crystal is shown in Figure 4a. The adhesion of the GO sheets to the surface of Co-BTC induces irregular particle sizes and surfaces (Figure 4b–d). Under low GO dosage, the GO sheets are disorderly dispersed on the Co-BTC surface, which are loosely stacked and connected to each other. With GO dosage increasing, the GO sheets are reciprocally interlaced and connected, leading to the full coverage of the Co-BTC surface, which is consistent with the SEM images. Because of the crumpled morphology of GO in the Co-BTC@GO composite (Figure 4b,c), the Co-BTC@GO composites are able to store more charges. In addition, the existence of Co-BTC could effectively prevent the aggregation of the GO sheets.

XPS measurement was used to determine the composition and valence state of the elements in the composites. Figure 5 shows the survey spectra of Co-BTC@GO, which mostly consists of C, O, and Co elements. The intense peaks at 287.64 and 532.68 eV are ascribed to C and O, respectively (Figure 5b,c) and correspond with previously published values [49,50]. As shown in Figure 5d, two peaks are found that have binding energies of 801.6 eV and 786.13 eV, which are ascribed to Co 2p1/2 and Co 2p3/2 [51,52]. Compared with Co-BTC, there are no obvious differences between the C1s peaks. As shown in Figure 5c,d, the binding energies of O1s, Co2p1, and Co2p3 in Co-BTC are 531.24, 800.17, and 785.48 eV, while in Co-BTC@GO-2, they equal 532.68, 801.05, and 786.13 eV, respectively. The binding energy of the O1s, Co 2p1, and Co2p3 peaks is stronger in the Co-BTC@ GO composites, indicating that interactions between Co-BTC and GO mainly occur between the -OH/-COOH in GO and Co^2+^ in Co-BTC.

### 3.2. Electrochemical Analysis

A three-electrode electrochemical system was applied to evaluate the charge and discharge performance of the synthesized Co-BTC@GO electrodes under different GO dosages in an electrolyte of 3 M KOH. Figure 6a shows the CV curves for the four electrodes at 2 mV/s in the potential window of 0–0.6 V. The two pairs of visible redox peaks indicate that the measured pseudo-capacitance is mainly determined by the reversible Faraday reaction of the Co^2+^ in the electrolyte solution [53,54]. All four materials exhibit redox and reduction reactions, in which the mechanism is attributed to the redox of Co-O with Co-O-OH, exhibiting typical pseudocapacitive behavior. Capacitance performance can be directly reflected by the integrated area of a CV curve. For the four electrolytes, the order of increasing integration area is Co-BTC@GO 4 < Co-BTC < Co-BTC@GO 1 < Co-BTC@GO 2, which is also representative of their specific capacitance order. The GCD curves of these four electrodes were further investigated.

The GCD measurement of the four as-prepared materials at 1 A/g is shown in Figure 6b. The specific capacitance is calculated based on the GCD curves by using Equation (1) in the Supporting Information [55].

The specific capacitances of Co-BTC, Co-BTC@GO 1, Co-BTC@GO 2, and Co-BTC@GO 4 are 759.2, 1036, 1144 and 741.1 F/g, respectively. This is attributed to the fact that the addition of GO effectively increases the electrical conductivity of Co-BTC by allowing more electron transport channels within the Co-BTC microspheres. In contrast, the electrochemical performance of Co-BTC@GO 4 is inferior to that of Co-BTC because of the incorporation of excess GO sheets in Co-BTC@GO 4, which leads to a significant increase in the aggregation effect of the nanoparticles. As shown in Figure 3 and Figure 4, we can observe that, for Co-BTC@GO 1 and Co-BTC@GO 2, the GO sheets are uniformly inserted into the MOF microspheres, increasing the conductivity of the composites and shortening the electron transport paths. However, for Co-BTC@GO 4, the excess GO sheets cover the surface of the MOF microspheres, blocking the pore channels of MOF and the transport paths of Co^2+^. The charging/discharging and capacitive behaviors of the composite electrodes prepared in this study may be related to the synergistic effects of GO, with Co-BTC GO providing sufficient space for the embedding and de-embedding of OH^−^. The three-dimensional structure with a porous microstructure, as well as high-speed charge transfer and exposure of active sites in the electrolyte, contribute to providing high specific capacitance.

The CV curves for Co-BTC@GO 2, which measure in the scan-rate range of 2–100 mV/s, are shown in Figure 6c (CV curves for Co-BTC, Co-BTC@GO 1, and Co-BTC@GO 4 are displayed in Appendix A). A more significant redox peak can be observed in the CV curve for Co-BTC@GO 2, indicating a substantially greater capacitance. Two sets of redox peaks can be observed in the CV curves, indicating that the capacitance is mainly a pseudo-capacitance provided for by the redox mechanism rather than a pure double layer capacitance. With the gradual increase in the scan rate, it can be seen that the reduction peak clearly shifts to the lower voltage, while the oxidation peak shifts to the higher voltage. The shape of the CV curves, on the other hand, remains consistent, indicating that the Co-BTC@GO 2 electrode has an excellent rate for properties and reversibility.

The GCD values for the Co-BTC@GO 2 electrode measure in the range of 1–20 A/g (Figure 6d) (GCD curves for Co-BTC, Co-BTC@GO 1, and Co-BTC@GO 4 are displayed in Appendix A). During the rapid charge-discharge process, Co-BTC@GO 2 shows excellent Coulombic efficiency, as shown by the symmetrical GCD curves. The specific capacitances calculated for Co-BTC@GO 2 were 1144, 1010.8, 944.4, 865, 744, and 620 F/g at 1, 2, 3, 5, 10, and 20 A/g, respectively. The specific capacitance of 1144 F/g at 1 A/g is the maximum value (with a comparison) (Table 1).

Figure 7a depicts the specific capacities of Co-BTC and Co-BTC@GO 2 at various current densities. The capacitance of Co-BTC@GO 2 remains at 744 F/g at 10 A/g, which is approximately 65% of the initial capacity, while those of Co-BTC, Co-BTC@GO 1, and Co-BTC@GO 4, are 63, 64, and 80%, respectively (Appendix A). The capacity of the Co-BTC@GO 2 electrode remains at 54.2%, even if the current density increases to 20 A/g. Clearly, the Co-BTC@GO 2 holds an excellent charge storage capacity and rate reversibility.

Figure 7b shows the capacitance of the Co-BTC@GO-hybrid supercapacitors, depending on the GO dosage. Among the electrodes tested, Co-BTC@GO 2 exhibits the best charge/discharge duration and excellent capacitance performance.

The electrochemical impedance spectrum (EIS) data were tested in the frequency range of 0.01–100 kHz, and the corresponding Nyquist plots were plotted based on the test results, as shown in Figure 7c. The Nyquist diagram of the tested material consists of two parts: the semicircular arc (high-frequency region) and the diagonal line (low-frequency region). The semicircular arc in the high-frequency region corresponds to the redox reaction at the electrode/electrolyte interface, while the diagonal line in the low-frequency region is closely related to the diffusion of the ions within the electrode [56,57]. To explain Figure 7c more intuitively, we built a simulated equivalent circuit with all fitting results for each parameter summarized in Table 2 [58].

The Rs and Rct of Co-BTC@GO 2 were 0.65 and 0.26 Ω, lower than that of Co-BTC (Rs = 1.07, Rct = 0.90 Ω). This indicates that the composite electrode has a faster charge transfer ability, and the OH^–^ ions in the electrolyte can diffuse to the electrode surface faster.

The Co-BTC@GO 2 electrode was evaluated for 2000 cycles in the potential range of 0–0.5 V at a current density of 20 A/g. As shown in Figure 7d, Co-BTC@GO 2 maintains 88.1% of its initial capacitance after 2000 cycles. However, after 2000 cycles, the capacitance retention of the Co-BTC electrode gradually decreased to 78.5%. This indicated that the Co-BTC@GO 2 composite electrode was more stable than the Co-BTC electrode. Due to the combination of MOF and graphene oxide, the volume change of MOF should reduce during OH^−^ insertion and de-insertion. Therefore, mixing metal-organic frameworks and graphene oxide can greatly improve cycling performance. By bonding Co-BTC@GO with graphene oxide, we have fabricated composites with excellent energy storage properties. As a result of this study, promising materials with excellent electrochemical properties for supercapacitors can be developed.

## 4. Conclusions

To enhance the charge capacitance capability of Co-BTC and avoid GO stacking via frequent strong directional interactions, Co-BTC@GO hybrid composites were architected and applied for use in supercapacitor electrodes. Microsphere-structured Co-BTC offers a framework effect for GO dispersion, contributing to a synergistic effect on the electrochemical performance of the resultant nanostructured composites.

Co-BTC@GO 2 has the highest specific capacitance, 1144 F/g. The Co-BTC@GO composite outperforms Co-BTC in terms of stereoscopic structure, electron transport resistance, and ion diffusion rate. Furthermore, after 2000 cycles, Co-BTC@GO 2 retained 88.1% of its initial capacity, compared to 78.5% for Co-BTC, indicating excellent cycle performance. The Co-BTC@GO composite electrode material is believed to have both the mechanical strength of graphene oxide with many functional groups and the controlled porous structure of MOF with organic ligands. This electrode material is expected to be applied to the fabrication of high-performance supercapacitors.

## Figures and Tables

**Figure 1 nanomaterials-12-03234-f001:**
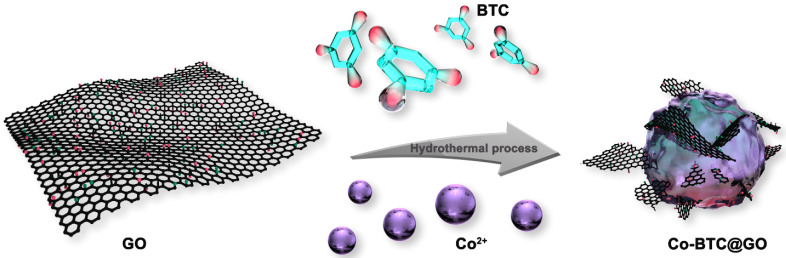
The preparation process of the Co-BTC@GO composites.

**Figure 2 nanomaterials-12-03234-f002:**
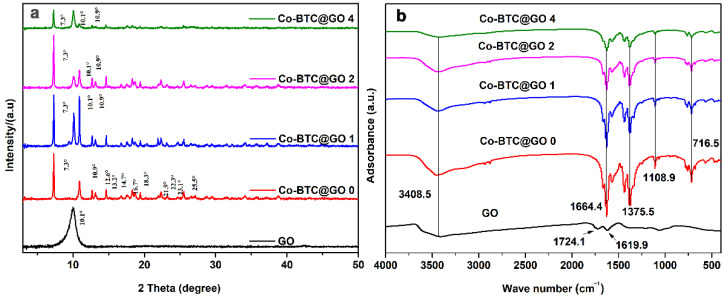
(**a**) XRD patterns for the GO, Co-BTC, and Co-BTC@GO composites; (**b**) FT-IR spectra of the GO, Co-BTC, and Co-BTC@GO composites.

**Figure 3 nanomaterials-12-03234-f003:**
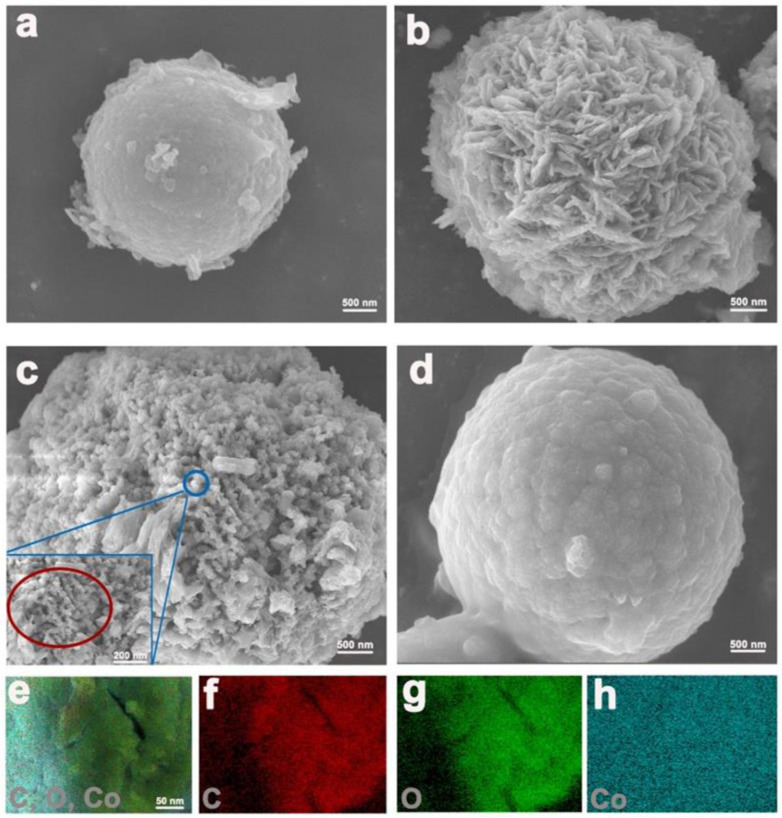
(**a**–**d**) SEM images of Co-BTC@GO 0, 1, 2, and 4; (**e**–**h**) EDS mapping of Co-BTC@GO.

**Figure 4 nanomaterials-12-03234-f004:**
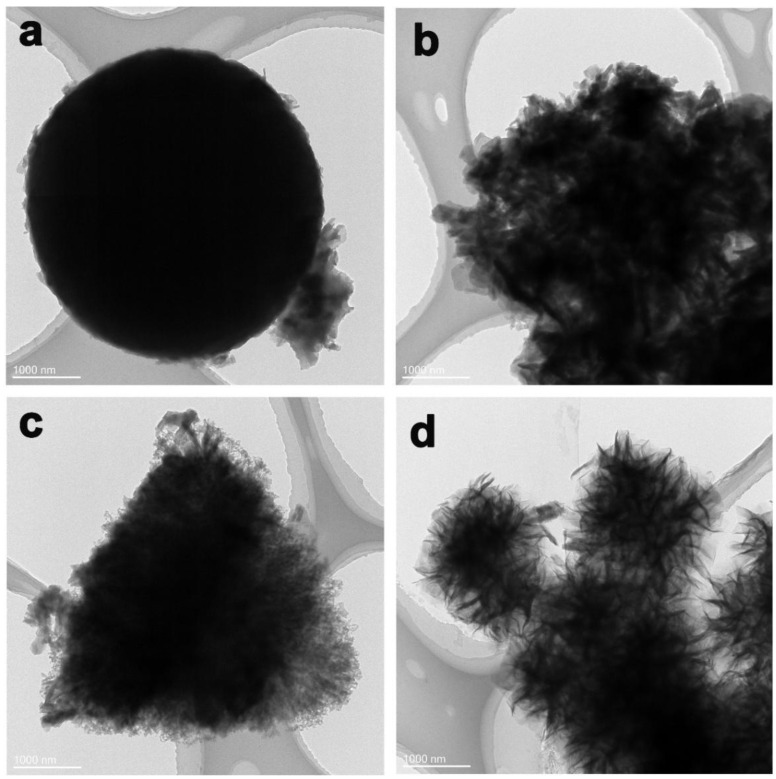
(**a**) TEM image of Co-BTC, (**b**–**d**) TEM images of Co-BTC@GO 1, 2 and 4.

**Figure 5 nanomaterials-12-03234-f005:**
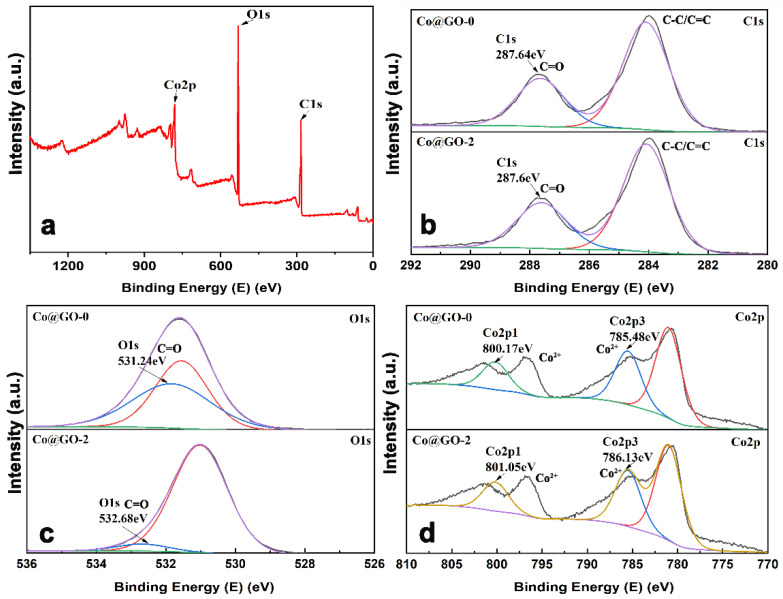
(**a**) XPS survey spectra of Co-BTC and Co-BTC@GO 2, (**b**) C1s, (**c**) O1s, (**d**) Co 2p.

**Figure 6 nanomaterials-12-03234-f006:**
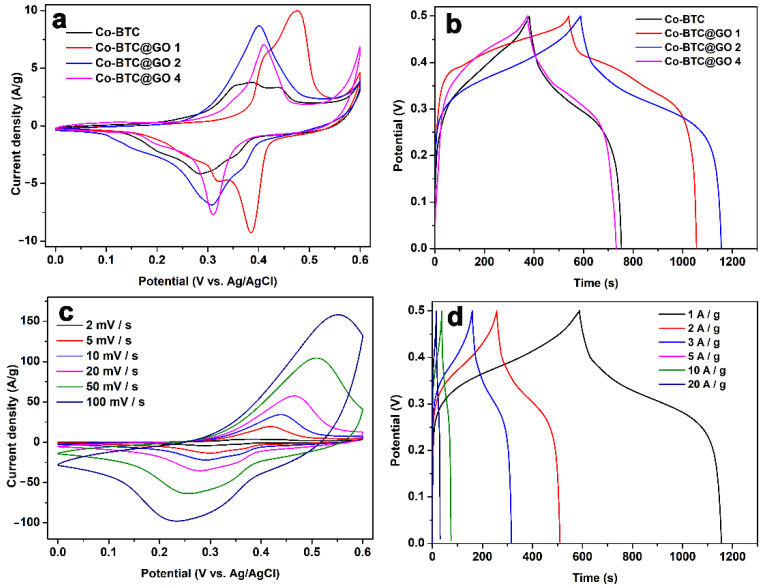
(**a**) CV curves for Co-BTC and Co-BTC@GO composites at 2 mV/s; (**b**) GCD curves for Co-BTC and Co-BTC@GO composites at 1 A/g; (**c**) CV curves for Co-BTC@GO 2 at 2–100 mV/s; (**d**) GCD curves for Co-BTC@GO 2 at 1–20 A/g.

**Figure 7 nanomaterials-12-03234-f007:**
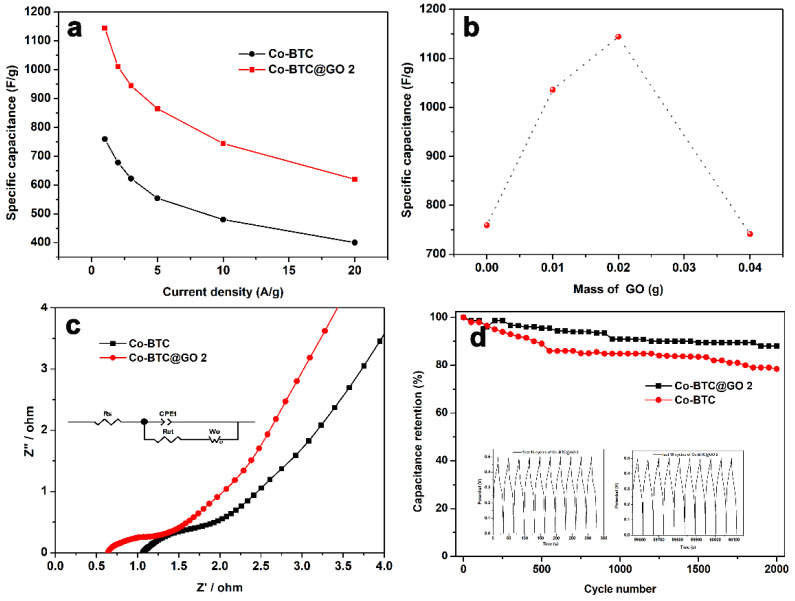
(**a**) The specific capacitance at various current densities for Co-BTC and Co-BTC@GO 2, (**b**) dependence of specific capacitance on the mass of GO at a current density of 1 A/g, (**c**) electrochemical impedance spectroscopy (EIS) (with a simulative equivalent circuit inside) of Co-BTC and Co-BTC@GO 2 in the frequency range of 0.01 Hz–100 kHz, (**d**) cycling performance (the inset is the GCD curves of the first cycles and last five cycles) of Co-BTC and Co-BTC@GO 2 at a current density of 20 A/g in 3 M KOH electrolyte.

**Table 1 nanomaterials-12-03234-t001:** Comparison of the electrochemical performance of similar MOF@GO composites reported in the literature.

Material	Capacitance	Scan Rate/Current Density	Electrolyte	Reference
CNTs@Co-BTC	553.3 F/g	1 A/g	1M LiOH	[39]
Ni-Co-MOF/GO	447.2 F/g	1 A/g	6M KOH	[41]
2DMOF/rGO film	292.5 F/g	0.7 A/g	1M H_2_SO_4_	[42]
2D/2D NiCo-MOF/GO	413.61 C/g	0.5 A/g	2M KOH	[44]
L-rGO-C-MOF	390 F/g	5 mV/s	1M NaNO_3_	[45]
Ni-BTC@GO 2	1144 F/g	1 A/g	3M KOH	This work

**Table 2 nanomaterials-12-03234-t002:** Equivalent circuit parameters obtained from EIS plots of Co-BTC and Co-BTC@GO 2.

	Rs (Ω)	Rct (Ω)
Co-BTC	1.07	0.90
Error (%)	1.3228	14.6740
Co-BTC@GO 2	0.65	0.26
Error	1.0077	14.5030

## Data Availability

Not applicable.

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
