# Peer review of "Architecting Nanostructured Co-BTC@GO Composites for Supercapacitor Electrode Application"

_nanomaterials, 2022, doi:10.3390/nano12183234_

Round 1

Reviewer 1 Report

The manuscript submitted by Chen et al. describes the preparation of a nanostructured Co-BTC@GO composite for supercapacitor electrode applications. It was shown that the preparation of a composite based on two known materials leads to a synergistic effect and to a noticeable improvement in the electrochemical properties important for applications. The optimal ratio of Co-BTC and GO has been experimentally determined.

The results obtained are new, of great interest and can be recommended for publication in Nanomaterials.

While reading the manuscript, the following questions and comments arose:

1)   Line 45 Metal organic frameworks (MOFs) NOT Metal organic frameworks (MOFs).

2)   The syntheses presented in the manuscript need a more detailed description. This will allow other researchers to confidently reproduce syntheses. It is indicated (line 90) that 23 ml of sulfuric acid was used. However, the concentration of the acid in the solution is not specified. It is also important to indicate the amount of substances obtained in syntheses.

3)   On Fig. 2 peak around 10o was correctly assigned to GO. However, in GO this peak is broad. It is necessary to explain why this peak is noticeably narrower in the Co-BTC@GO-1 composite.

4)   Line 124: “The excessive addition of GO, therefore, has a great effect on the crystal structure of Co-BTC”. It would be fine if the authors can explain what changes are taking place in the structure of Co-BTC.

Reviewer 2 Report

In this work Co-BTC@GO composites have been synthetized and tested as supercapacitor electrodes. The proposed work plan is correct to me, but the some improvements are needed. Major revisions are requested. A linguistic revision is also needed to remove grammar and typo errors, some sentences are not clear and need to be rewritten. In the following a list of items to be addressed during the revision process.

Introduction

-          Remove details about the results obtained from the last part and leave there only the aim of the work.

Experimental section

-          Details on the analytical techniques used and the conditions applied to characterize the synthetized materials have to be reported

-          Why the volume of DFM used during the preparation of composites is different from that used during the synthesis of pure Co-BTC?

Results and discussion section

-          The description of IR bands ascribable to BTC is too summary, please improve it

-          Lines 148-153: please revise this part, it is not clear.

-          The morphology of Co-BTC GO 4 is very similar to that of Co-BTC, can the Authors comment on this result?

-          Lines 173-177: what does it mean the part of the sentence: “that the binding energy of O1s, Co2p1 and Co2p3 peaks is stronger”?, please revise it.

-          Lines 197-202: please revise this part, it is not clear.

-          Section 3.2: I suggest to provide a comparison between the achieved results whit those already published on electrodes made by MOF-GO composites

Reviewer 3 Report

#Reviewer 1

The present work by Tianen Chen et al. reported an innovative graphene oxide (GO)-induced strategy for synthesizing GO-based metal-organic framework composites (Co-BTC@GO) supercapacitor electrode materials. The Nano architect Co-BTC@GO composites show good specific capacitance performance. This work is interesting and will provide some information on how to create microsphere-like MOF composites with carbonaceous materials as efficient electrode materials for supercapacitors. Therefore, I suggest this research paper can be accepted after some revisions as listed below:

1. The scales of Figure 3 (f, g, and h) in the manuscript are not mention and the authors are advised to mention.

2. The numbering pattern is not uniform in figure S3 and the authors are advised to make uniform.

3. In the analysis of XRD on page 13, some additional peaks are also seen in Co-BTC@GO1, and Co-BTC@GO 2, and the intensity of peaks are almost equal in Co-BTC, Co-BTC@GO 1, and Co-BTC@GO 2, but the intensity of peaks in Co-BTC@GO 4 is reduced. The authors are advised to mention the reason behind this? 

4. It is recommended to supplement the relevant data of cyclic GCD stability test after 20,00 cycles of electrode materials.

5. The mass of active materials used during electrode fabrication is not mention. So, the authors are advised to mention this? 

6.The redox mechanism is not clearly mention in the manuscript. So, the authors are advised to mention clearly.

 7.The authors may need to cite more relevant literatures to enrich their introduction, such as 10.1016/j.cej.2022.138363; 10.3390/inorganics10060086;

Round 2

Reviewer 2 Report

The Authors revised the manuscript in accordance with my comments, but some improvements are still needed. Minor revisions are requested. A linguistic revision has to be performed to remove grammar and typo errors. The correction of the IR bands description is not readable; I suggest to make a check of such part. I recommend to move from the supporting information to the main text the information about the instrumental techniques and the methods used to characterize the materials.

Author Response

Point 1: A linguistic revision has to be performed to remove grammar and typo errors.

Response 1: Thanks for your suggestion. I have read through the entire text and found and corrected some grammatical errors and typos—for example, line 24, line 46 and so on.

Point 2: The correction of the IR bands description is not readable; I suggest to make a check of such part.

Response 2: Thanks for the suggestion. Descriptions about the IR bands are adjusted according to the reviewer’s comment (Page 4, lines 152-156). Please check it.

Point 2: I recommend to move from the supporting information to the main text the information about the instrumental techniques and the methods used to characterize the materials.

Response 3: Thanks for the suggestion. The information about the instrumental techniques and the methods used to characterize the materials has moved to the main manuscript (Page 3, lines 120-131).

Reviewer 3 Report

The article is improved for publication in this journal.

Author Response

Thank you so much for your constructive advice. It has improved my article a lot and I have learned a lot from it.